# Characterization of Volatile Organic Compounds in Kiwiberries (*Actinidia arguta*) Exposed to High Hydrostatic Pressure Processing by HS-SPME/GC-MS

**DOI:** 10.3390/molecules27185914

**Published:** 2022-09-12

**Authors:** Małgorzata Starowicz, Wioletta Błaszczak, Ewa Ciska, Piotr Latocha

**Affiliations:** 1Institute of Animal Reproduction and Food Research, Polish Academy of Sciences, Tuwima 10, 10-748 Olsztyn, Poland; 2Institute of Horticulture Sciences, Warsaw University of Life Sciences—SGGW, Nowoursynowska 166, 02-787 Warsaw, Poland

**Keywords:** cultivars, kiwiberry, high hydrostatic pressure, volatiles, microextraction, GC-MS

## Abstract

HS-SPME/GC-MS analysis was carried out to characterize the profile of volatile organic compounds (VOCs) in kiwiberry cultivars (Geneva and Weiki) exposed to high hydrostatic pressure (HHP) (450–550–650/5 and 15 min). The sum of individual VOCs in Geneva (6.493 mg/kg) and Weiki (11.939 mg/kg) samples was found to be significantly reduced after processing, particularly for pressurization conditions of 650 MPa/15 min (decrease of 62%) and 550 MPa/15 min (decrease of 84%), respectively. On the other hand, Geneva and Weiki exposed to 450 MPa/5 min manifested the lowest loss in the sum of the VOCs. Geneva exposure to 450 MPa/5 min led to an increase in the hexanal (r = 0.782) and linalool (r = 0.806) content. Sample pressurization (450 MPa/15 min) promoted the formation of methyl butanoate, ethyl hexanoate, and *cis*-geraniol, simultaneously increasing the benzaldehyde (r = 0.886) concentration. However, the treatment of Weiki at 450 MPa/5 min favored trans-2-heptenal (r = 0.999) and linalool (r = 0.970) formation, as well as the (-)-terpinen-4-ol (r = 0.848) and geraniol (r = 0.694) content. Ethyl butanoate, hexanal, and 1-octen-3-ol were highly concentrated in the HHP-treated (450 MPa/5 or 15 min) Weiki. Pressurization decreased the terpenoid contribution, but also increased the contribution of alcohols and aldehydes to the overall VOC number in both tested cultivars.

## 1. Introduction

Kiwiberry (*Actinidia arguta*) is included in the group of dessert fruit, and it is grown in many European countries, including Poland. Kiwiberry contains over 20 essential nutrients, making it one of the most nutrient-dense fruits [1]. Kiwiberry varies from the well-known and popular fruit of the *Actinidia deliciosa* or *Actinidia chinensis*, since it is smaller and hairless. The kiwiberry is an attractive fruit for consumers, due to its health-promoting potential [2] and rich aromatic flavor [3]. This fruit has been shown to have tropical fruit, candy, and green aromas. However, the aroma of the common kiwifruit has been described as more green, grassy, and sulfurous, with a melon and sweet candy flavor [3,4]. The volatile organic compounds (VOCs) of kiwiberries are mainly esters, aldehydes, alcohols, and some monoterpenes.

The delicate fruit of kiwiberry very often responds to improper storage conditions with changes in fruit quality, such as skin softening and wrinkling. Moreover, during kiwiberry production, high amounts of damaged fruit, or fruit without the proper size, are generated. The fruit that does not meet the standards required for the dessert fruit cannot be commercialized; therefore, it is frequently processed into jams and juices [2].

It is well known that plant cultivars, ripening, storage, and technological processes are the key factors affecting the profile of VOCs in kiwifruit-based products [3]. As reported by these authors, kiwifruit-based products demonstrate strong off-odors, which have been described as old cut grass, hay, and overripe. The processing of kiwifruit juice favored the formation of (E)-3-hexenal, which is responsible for the characteristic hay off-odor. In contrast, the aroma of kiwiberry juice—not from concentrate—has been described as green, grassy, fruity, floral, resulting from a propenal, hexanal, pentane, 1-hexanol, and methanol composition [5]. Overripe and terpene-like off-flavors were formed in kiwifruit puree upon storage, which were related to the formation of VOCs such as (*E*)-2-hexenal, cyclohexanone, terpene esters, linalyl acetate, isobornyl acetate, and α-terpenyl acetate. On the other hand, the main odorants found in fresh puree are 3-penten-2-ol, ethyl butanoate, (*E*)-2-hexenal, 6-methyl-5-hepten-2-one, 1-octen-3-ol, methyl benzoate, and hexyl hexanoate [6]. Among the freshness descriptors for fruits, aroma was a principal factor for high consumer acceptance [7]. Traditional thermal processes increased the stability, and thus, the shelf life, of fruit-based products, but distinctly reduced the fresh- and fruity-like aroma notes.

High hydrostatic pressure (HHP) is a physical process used in food preservation, wherein the food is exposed to pressure (up to 700 MPa), and the processing temperature usually does not exceed 50 °C. Therefore, HHP enabled the production of products with prolonged shelf lives and with high nutritional quality due to the preservation of the majority of bioactive food constituents [8]. Moreover, HHP-treated plant-based products demonstrated a natural fresh-like appearance and aroma [9]. According to Zhao et al. [9], the juice obtained from the common kiwifruit and exposed to a pressure of 500 MPa for 10 min presents a similar volatile profile to that of a fresh sample. Chen et al. [10] demonstrated that ethyl acetate, with ethereal, fruity, and brandy-like aromas found in fresh kiwifruit pulp, does not change after HHP processing (400–500–600 MPa/5–10–15 min). In contrast, a substantial decrease in the butyl acetate content (ester with fruity aroma notes) was observed after sample exposure to pressures above 400 MPa.

The influence of high-pressure treatment on the VOCs in the common kiwifruit of *A. deliciosa* or *A. chinensis* had been evaluated in previous reports [9,10,11]. However, there are scarce data available regarding the effect of pressurization on the profile of volatile compounds in kiwiberries (*A. arguta*) with an emphasis on the plant cultivars. Therefore, the aim of this research was to study the profile of VOCs in two kiwiberry cultivars (cvs), i.e., Geneva and Weiki, as well as to characterize the changes in the VOC profiles as a result of kiwiberry processing under HHP (450–550–650 MPa for 5 and 15 min). We elaborated the VOC profile using headspace solid-phase microextraction (HS-SPME) with gas chromatography-mass spectrometry (GC-MS). Additionally, statistical tools were used to determine the relationship between the studied parameters.

## 2. Results and Discussion

### 2.1. Qualitative Analysis of VOCs in Kiwiberry Cultivars

The cv Geneva used in this study had fruit with green flesh and skin, while the fruit of cv Weiki was characterized by green flesh and a red blush. The tested cvs showed significant variation in the profile of the volatile organic compounds (VOCs). A total of 39 and 68 volatile organic compounds (VOCs) were detected in all the tested (untreated and HHP-treated) samples of cvs Geneva and Weiki, respectively (Table 1 and Table 2). Most of the VOCs were linked with their aroma notes. Therefore, aroma descriptors for identified compounds were collected from the online databases pherobase.com and pubchem.com (accessed on 1 August 2022).

A total of 18 and 28 VOCs were identified in the untreated Geneva (Table 1) and Weiki (Table 2) kiwiberries, respectively. However, the HHP-treated samples showed a different number of VOCs, depending on the parameters used upon HHP processing (450, 550, 650 MPa, for 5 and 15 min).

As shown, pressurization of cv Geneva led to a distinct increase in the number of identified VOCs. The highest number of VOCs (30 highly volatilized compounds) was found after sample processing at 450 MPa for 15 min. However, the lowest VOC number (22 compounds) was distinguished after sample exposure to 450 MPa/5 min. Evidently, the pressurization of Geneva favored the formation of ethyl hexanoate, octanal, 3-octanol, 3,5,5-trimethyl-2-hexene, *p*-α-dimethyl-styrene, (*E*)-2-octenal, 2-ethyl-1-hexanol, (*E*,*E*)-2,4-heptadienal, 1-nonanol, *cis*-geraniol, or *p*-metha-1,8-dien-9-ol. However, methyl butanoate, (*Z*)-2-decanal, *o*-cymene, ethyl caprylate, and octanoic acid were only observed after sample exposure to 450 MPa/15 min. In contrast, α-myrcene and carvone are lost after processing.

Similarly, the HHP treatment of cv Weiki increased the number of VOCs in the tested samples (Table 2). The highest number of VOCs (45 volatiles) was identified after Weiki exposure to 450 MPa for 5 min. Pentanal, hexyl acetate, octanal, (*Z*)-3-hexen-1-ol acetate, (*E*)-2-hexen-1-ol acetate, 1-hexanol, 3-heksen-1-ol, 3-octanol, (*E*)-2-heksen-1-ol, 3,5,5-trimethyl-2-hexene, and α-linalool were detected in the HHP-treated samples. Some of the volatilized compounds were formed only under particular conditions for the HHP process; for example, the treatment of Weiki at 450 MPa/5 min led to the formation of 9 new compounds: α-terpinene, (2*Z*)-3-pentyl-2,4-pentadien-1-ol, (*Z*)-2-heptenal, (*E*,*E*)-2,4-heptadienal, (*Z*)-2-decanal, verbenol, 1-nonanol, *cis*-citral, and carvone. On the other hand, 2-heptanone, isopentyl alcohol, 3-hydroxy-2-butanone, 6-methyl-3-heptanol, ethyl caprylate, 2-ethyl-1-hexanol, benzaldehyde, and octanoic acid were generated under 650 MPa for 5 and/or 15 min. In contrast, *E*,*E*-2,6-dimethyl-1,3,5,7-octatetraene, butanoic acid, and *cis*-carveol were lost after pressurization.

The major volatile compounds of *A. arguta* were mainly esters, alcohols, and aldehydes, with ethyl butanoate, ethyl hexanoate, (*E*)-2-hexenal, and hexanal as the main contributors to its aroma [12]. These authors also reported that 1-penten-3-one, pentanal, 1-octen-3-ol, linalool, terpinen-4-ol, and α-terpineol substantially contribute to the kiwiberry aroma of the cv Ananasnaya. Moreover, flavor compounds also accumulate in plant tissue material as nonvolatile glycoconjugates, and they have been considered an important potential source of flavor compounds, which can be released from the sugar moiety during either maturation or processing [3]. According to the findings presented in the work of Garcia et al. [12], hexanal and 1-octen-3-ol appear in both the bound and free volatile extracts of ripe *A. arguta*. It is well known that HHP induces the disruption of plant cell walls through mechanical damage during compression (pressure build up) and decompression phenomena, which in turn can facilitate both the extractability and release of some compounds from the complexes formed with other plant tissue constituents [2]. This phenomenon was likely responsible for the appearance of new volatiles in the group of samples subjected to the HHP treatment.

VOCs identified in the cvs Geneva and Weiki can be grouped into different chemical classes: alcohols, aldehydes, carboxylic acids, esters, ketones, and terpenoids (Figure 1 and Figure 2). The predominant group identified in the native Geneva (Figure 1) are terpenoids (39% of the total number of identified VOCs), followed by aldehydes (22%), esters (17%), alcohols (11%), and ketones (11%). Evidently, the contribution of terpenoids to the VOC profile of Geneva is reduced after HHP treatment. The largest decrease in this contribution (from 39% to 19%) was observed after sample exposure to 650 MPa (irrespective of the processing time). In contrast, a distinct increase was found in the contribution of the alcohols (from 11 to 27%) and aldehydes (from 22 to 31%) to the VOC profile after sample processing at 650 MPa for 5 min and 15 min, respectively. The results presented were consistent with recently published data [9], which showed an increasing trend towards the contribution of alcohols to the VOC profile of kiwifruit juice after processing at 500 MPa for 10 min. Contrary to our results, however, these authors observed a slight reduction in the aldehyde contribution after pressurization, which in turn resulted in a deterioration in the freshness of kiwifruit juice, as aldehydes are mainly desirable compounds, with a grassy aroma.

Similarly, the dominant volatile group found in the untreated Weiki were terpenoids (Figure 2), and in this case, pressurization decreased their contribution to the VOC profile. The most pronounced drop (from 52% to 41%) was observed after sample exposure to 450 MPa/15 min, 550 MPa/5 min, and 650 MPa/15 min, respectively. On the other hand, the pressurization of Weiki induced an increase in the contribution of aldehydes and alcohols. The most distinct increased in aldehyde (from 7% to 18%) and alcohol (from 11% to 19%) contributions was found after exposure of Weiki to a pressure of 450 MPa for 5 and 15 min, respectively. As mentioned above, the contribution of esters showed an irregular change with pressurization. Exposure of Weiki to higher pressure levels distinctly increased the contribution of ketones, particularly the treatment at 650 MPa/15 min, after which an increase from 4% to 10% was observed.

As mentioned above, the profile of the VOCs, with an emphasis on hydrocarbon terpenes, alcohols, esters, or aldehydes, can be altered by HHP processing. Moreover, pressurization can enhance and/or retard various enzymatic and chemical reactions, and this, in turn, can directly affect the content of particular volatiles, as well as their contribution to the kiwiberry aroma [13]. According to García-Parra et al. [14], most volatile compounds originate from lipid oxidation, carotenoids degradation, and/or Maillard reactions. As reported, peroxidase (POD) promotes lipid oxidation, with consequent off-flavor formation, and PPO is responsible for enzymatic browning [15]. Therefore, it can be concluded that the changes observed in volatile composition can also be due to the POD/PPO activity. It was shown that effective inactivation of POD isolated from kiwifruit required a pressure higher than 400 MPa, combined with temperature treatment at 50 °C, as well as processing time of up to 30 min [16].

### 2.2. Quantitative Analysis of VOCs in Geneva and Weiki cvs

A total of 12 and 18 major volatiles of untreated Geneva and Weiki samples analyzed by GC-MS were tabulated in Table 3 and Table 4, respectively.

As indicated, the concentration of the most important volatiles (ethyl butanoate, ethyl hexanoate, hexanal, 1-octen-3-ol) to the aroma of *A. arguta* [3] varies distinctly between the studied cv Geneva and cv Weiki. A nearly three-fold higher concentration of ethyl butanoate was found in the untreated Weiki kiwiberry than in cv Geneva. Similarly, ethyl hexanoate, which was not found in the untreated cv Geneva, is the second predominant ester in the Weiki sample. In contrast, other green-smelling esters, such as methyl benzoate and ethyl benzoate, were more dominant in the Geneva sample than in the Weiki sample. Geneva manifests a more than three-fold higher concentration of hexanal than Weiki. However, 1-octen-3-ol was not found in Geneva. It is worth mentioning that terpinolene was the major volatile in the untreated Weiki. Although terpinolene was also found to be the predominant volatile in Geneva, its concentration was 72% lower compared to the concentration noted in the Weiki sample. The untreated Weiki manifested a high concentration of eucalyptol—the minty, sweet terpenoid—which was not found in the cv Geneva. As indicated, eucalyptol has been identified in yellow-fleshed (Gold) kiwifruit, in relatively high concentrations [17].

The results obtained from GC-MS analysis prove that the HHP treatment significantly affected the VOC profile in the tested Geneva samples (Table 3). The sum of VOCs in Geneva (6.493 mg/kg) was found to be reduced after processing, particularly for pressurization conditions of 650 MPa/15 min (a decrease of 62%). On the other hand, Geneva exposed to 450 MPa/5 min manifested the lowest loss in the sum of the VOCs. The described changes of VOCs could be easily observed in superimposed chromatograms of untreated and HPP-treated Geneva samples (Appendix A). As shown, the concentration of hexanal increased significantly (nearly 40%) after sample treatment at 450 MPa/5 min compared to the untreated Geneva. However, further processing of Geneva resulted in hexanal reduction, and the most pronounced decrease (82%) was observed after processing at 550 MPa/15 min. Hexanal, the product of lipid degradation, was found to be the predominant volatile of kiwifruit [9]. Indeed, despite the observed decrease, the hexanal and 1-hexanol content were found to be the highest compared to the estimated concentration of the other VOCs in the Geneva samples. However, in contrast to hexanal, the concentration of 1-hexanol increased significantly throughout the sample processing, and the most distinct increase (82%) was found after exposure of the sample to 550 MPa/5 min. According to Zhao et al. [9], both of these volatiles were responsible for the characteristic fresh and grassy kiwifruit aroma. Similarly, the content of benzaldehyde increased after the processing of the Geneva sample, with an emphasis on the sample exposed to 450 MPa/15 min, where a nearly five-fold increase was noted. This volatile has been previously found in Hayward kiwifruit (*A. deliciosa*) as the major bound volatile compound. A high concentration of other benzaldehydes has been reported to be responsible for the apple aroma of *A. deliciosa* [3]. We also noted a more than 20% increase in the linalool concentration after Geneva processing at 450 MPa/5 min. Moreover, processing at 450 MPa for 15 min favored the formation of methyl butanoate and ethyl hexanoate. Similarly, the tested processing parameters led to the formation of *cis*-geraniol, with the highest concentration obtained after sample treatment at 550 MPa/15 min. As mentioned above, aldehydes were responsible for the freshness of kiwifruit, while terpenoids are floral volatiles [3,9].

The sum of VOCs in Weiki (11.939 mg/kg) samples was found to be significantly reduced after processing, particularly for pressurization conditions of 550 MPa/15 min (decrease of 84%). The smallest change in the sum of the VOCs was noticed in Weiki exposed to 450 MPa/5 min. The 1-octen-3-ol was another predominant aroma contributor of *A. arguta* [12] (Table 4). This volatile was characterized by a highly aromatic, sweet, and earthy odor, with a strong herbaceous note [18]. The HHP treatment of Weiki favored the formation of 1-hexanol. Similarly, pressurization led to the formation of linalool and benzaldehyde. However, the latter volatile was observed only in the samples treated at 650 MPa. The concentration of geraniol was also increased by nearly 30% after processing at 450 MPa/5 min. As mentioned above, linalool and geraniol were floral terpenoids found in kiwifruit, mainly in bound form [3,12]. In contrast, the HHP treatment of both Geneva and Weiki kiwiberries resulted in a distinct decrease in ethyl butanoate concentration. As reported, ethyl butanoate was the most common ester of kiwiberry fruit [19] and has been described as the major contributor to the kiwiberry aroma [20].

Moreover, not only ethyl butanoate, but also methyl benzoate and ethyl benzoate were identified (by GC-MS-O) to be present in the free volatile extracts of *A. arguta*, with highly distinct odor intensities [11].

Considering the results obtained in our study, as well as in the above-cited reports, we can conclude that volatiles such as 1-octen-3-ol, benzaldehyde, hexanal, linalool, and *cis*-geraniol can be effectively released from the sugar moiety for certain parameters of the pressurization process for the tested kiwiberry cultivars. However, volatiles classified in the literature as free-form compounds, with an emphasis on ethyl butanoate and methyl butanoate, appeared to be more vulnerable to the pressurization process. We can also conclude that as for the case of Geneva, the HHP treatment of Weiki also reduced the sum of the individual VOCs. However, for the case of Weiki, the most pronounced decrease (by 80%) was noted after exposure to 550 MPa/15 min. Following the assumptions of Oey et al. [12], we may conclude that the effect of HHP on the VOCs profile is rather complex. The observed decrease in the concentration of volatile compounds likely resulted from a fact that pressurization enhanced and/or retarded some enzymatic (i.e., PPO/POD) activity, as well as chemical reactions, such as oxidative chemical reactions or hydrolysis [10], which are the key factors for the stability of VOCs. Therefore, not only was the content of the individual volatiles disturbed, but the entire profile of the VOCs was also changed as a consequence of the HHP processing. Our preliminary results (not shown in this article) indicated that the conditions used during HHP treatment (450–550–650 MPa/5 and 15 min) of the kiwiberry samples were insufficient to completely inhibit PPO and POD activity. On the other hand, they were equally effective in inhibiting the growth of microorganisms to a level below 10 microorganisms per 1 g of the FW of kiwiberry sample. Therefore, we can confidently state that the effect of microorganisms and the processes resulting from their activity on the profile of the tested VOCs can be excluded.

Multivariate PLS-DA analysis was performed for untreated and HHP-treated (450–550–650 MPa/5 and 15 min) samples and different variables (individual VOC concentration, sum of individual VOCs in the tested material) to investigate the structure and regularity of the relationships between variables for the cases of Geneva (Figure 3) or Weiki (Figure 4).

As indicated by the obtained results, the VOC profile of the untreated Geneva sample (Figure 3) was strongly correlated with the ethyl benzoate (r = 0,987), 6-methyl-5-hepten-2-one (r = 0.975), terpinolene (r = 0.974), methyl benzoate (r = 0.881), ethyl butanoate (r = 0.872), and (R)-(+)-limonene (r = 0.815) content. The sample treated at 450 MPa/5 was highly and positively correlated with linalool (r = 0.806) and hexanal (r = 0.782). The concentrations of methyl benzoate (r = 0.986), ethyl benzoate (r = 0.930), and (R)-(+)-limonene (r = 0.978) were significantly correlated with the sum of the individual VOCs in the tested Geneva sample. Similarly, terpinolene (r = 0.892), 6-methyl-5-hepten-2-one (r = 0.848), and ethyl butanoate (r = 0.791) were positively correlated with the sum of the individual VOCs. The HHP treatment (450 MPa/15 min) of Geneva was highly correlated with the benzaldehyde (r = 0.886) and ethyl hexanoate (r = 0.989) concentrations. Moreover, the sample treated at 550 MPa/15 min was significantly correlated with the *cis*-geraniol content (r = 0.582), while the sample exposed to 550 MPa/5 min was well correlated with 1-hexanol (r = 0.796).

The VOC profile of the untreated Weiki (Figure 4) was characterized by a significant contents of *α*-pinene (r = 0.972), ethyl benzoate (r = 0.998), methyl benzoate (r = 0.942), *γ*-terpinene (r = 0.948), eucalyptol (r = 0.932), ethyl butanoate (r = 0.868), and 6-methyl-5-hepten-2-one (r = 0.876). However, the VOC profile for Weiki exposed to 450 MPa/5 min was significantly correlated with the concentrations of trans-2-heptenal (r = 0.999) and linalool (r = 0.970) and with the (-)-terpinen-4-ol and *cis*-geraniol (r = 0.848 and 0.694, respectively) contents. The benzaldehyde concentration was positively correlated only with the Weiki processing at 650 MPa/5 and 15 min (r = 0.625). The content of 1-hexanol was well correlated with the Weiki treatment at 650 MPa/15 min (r = 0.416), while the 1-octen-3-ol concentration was correlated with the sample exposure at 450 MPa/15 min and 650 MPa/5 min (r = 0.540). The hexanal content was positively correlated with the sample treatment at 450 MPa/15 min (r = 0.590) and negatively correlated with the exposure to 550 MPa/15 min (r = −0.798). The sample processing at 550 MPa/15 min was correlated with the ethyl hexanoate content (r = 0.864).

## 3. Materials and Methods

### 3.1. Chemicals

The VOCs standards *β*-*cis*-ocimene, benzaldehyde (≥99%), *cis*-geraniol (≥97%), hexanal (≥95%), 1-hexanol (≥98%), ethyl benzoate, ethyl butanoate, ethyl hexanoate, ethyl octanoate, eucalyptol (≥99%), methyl benzoate, methyl butanoate (≥98%), 6-methyl-5-hepten-2-one, myrcene (≥90%), (R)-(+)-limonene (~90%), linalool (≥97%), *α*-pinene, 1-octen-3-ol (≥95%), *γ*-terpinene, terpinolene, (-)-terpinen-4-ol, trans-2-heptenal (≥95%), and C_7_-C_40_ n-alkanes mix were purchased from Sigma (Sigma-Aldrich, Saint Louis, MO, USA). All standards were of at least analytical grade.

### 3.2. Material and Sample Preparation

The kiwiberry cv Geneva and Weiki were purchased from a kiwiberry orchard, Mini Kiwi Kostrzewa (Bodzew, Belsk Duży, Poland). The obtained healthy and fresh fruit was kept at 8 °C until it reached the eating-ripe stage of maturity. The fruit taken for the analysis (~1000 g of each cultivar) was characterized by similar average values for the titratable acidity and soluble solid content, i.e.:-0.90 g/100 g citric acid and 19 °Brix, respectively, for Geneva;-1.23 g/100 g citric acid and 18.5 °Brix, respectively, for Weiki.

After removing the peduncle and stem-ends, the fruit (~150 g) was homogenized using a laboratory mixer (B-400, Buchi, Switzerland) equipped with ceramic knives. The homogenization process was carried out at 9000 rpm, and homogenization time was not longer than 15 s. The procedure was performed in triplicate for each cultivar, and the sample obtained was immediately subjected to HHP processing.

### 3.3. High Hydrostatic Pressure (HHP) Processing

Pressurization was carried out according to the conditions and parameters used in our previous studies for kiwiberry fruit [2]. Briefly, Teflon tubes (50 mL) filled with homogenized fruit were closed and placed in a high-hydrostatic pressure chamber (100 mL, High Pressure System U33, Unipress, Warsaw, Poland). The chamber was filled with a pressure-transmitting medium (water—propylene glycol (propane-1,2-diol), 1:1, *v*/*v*), and the samples were exposed to pressures of 450 MPa, 550 MPa, and 650 MPa for 5 and 15 min. The processing temperatures were 33 ± 1.5 °C (for 450 MPa), 36 ± 1.5 °C (for 550 MPa), and 39 ± 1.5 °C (for 650 MPa), and the compression and decompression rates were 8 MPa/s and 10 MPa/s, respectively. The processing parameters were monitored using OMEGASOFT^®^ Data Acquisition Software (OMB DAQ-54, OMEGA Inc., Budapest, Hungary). The analyses were performed in triplicate for each cultivar and production step. The untreated and HHP-treated samples were stored at −24 °C (for no longer than 7 days) until analysis.

### 3.4. Headspace Solid-Phase Microextraction (HS-SPME)

Volatile compounds of the kiwifruits were extracted based on the HS-SPME technique, according to Starowicz et al. [21] and Wan et al. [22], with slight modifications. The HS-SPME parameters were established in the initial experiment (data not shown). Therefore, 2 g of the thawed material was weighed into a 20 mL vial and sealed with a PTFE-silicon septum. The vials were placed onto an Eppendorf agitator/heater (Eppendorf, Hamburg, Germany), shaken and heated (40 °C, 50 min), and then the volatiles were allowed to absorb onto the SPME fiber for 15 min at 50 °C. A divinylbenzene/carboxen/polydimethylsiloxane (DVB/CAR/PDMS) fiber (2 cm, 50/30 µm, stable flex, Supelco, Bellefonte, PA, USA) was used for extraction, and the fiber was installed into the manual SPME holder. According to the producer’s instructions, DVB/CAR/PDMS fiber was conditioned for 1 h at 270 °C in a GC injector port (Agilent Technologies 7890A GC system, Santa Clara, CA, USA). The injection was performed manually. Thus, the SPME fiber was introduced into the chromatograph injection port (splitless mode), where the analytes were desorbed at 250 °C for 5 min.

### 3.5. Gas Chromatography–Mass Spectrometry (GC-MS) Analysis

The GC-MS analyses were performed using a gas chromatograph (Agilent Technologies 7890A GC system, Santa Clara, CA, USA) coupled to a mass spectrometer (Agilent Technologies 5975C VL MSD, Santa Clara, CA, USA). The volatiles were separated onto a capillary column (DB-WAX, 30 m, 0.25 mm × 0.50 µm). The temperature was initially set to 40 °C and held for 5 min. The temperature was then increased to 200 °C and held for 1 min. Finally, the temperature was increased to 240 °C and held for 5 min. In this method, helium was used as the carrier gas, with a flow rate of 1 mL min^−1^. The flow was kept constant. The electron ionization source temperature was 230 °C, and the ionization electron energy was 70 eV. The mass range was scanned in the full-scan acquisition mode from 30–450.

The volatile compounds were identified by comparing the obtained linear retention indices, retention times, and mass spectra with the Wiley Registry 7th Edition Mass Spectral Library (Wiley and Sons Inc., Weinheim, Germany) and the National Institute Standards and Technology (NIST) 2005 Mass Spectral Library. Linear retention indices (LRIs) were elaborated using a C_7_-C_40_ n-alkanes mix (Sigma-Aldrich, Saint Louis, MO, USA). The results were calculated based on the calibration curves for the external standards (R^2^ = 0.980–0.998). All analyses were carried out in triplicate.

### 3.6. Statistical Analysis

The obtained data were analyzed using one-way ANOVA, followed by Tukey’s test (*p* ≤ 0.05) for post hoc multiple comparison of the means. Untreated and HHP-treated (450–550–650 MPa/5 and 15 min) sample comparisons were performed to assess the structure and regularity in the relationships between variables. Thus, partial least squares discrimination analysis (PLS-DA) was applied to study the correlations that may be apparent between the HPP treatment and VOCs for the cases of Geneva (Figure 3) or Weiki (Figure 4). The statistical study was implemented using Statistica 13.1 (Statsoft Inc., Tulsa, OH, USA) and Simca 17.0 (Sartorius, Göttingen, Deutschland).

## 4. Conclusions

HS-SPME/GC-MS analysis of the volatile profile of kiwiberry revealed significant differences in the number, contribution, and concentration of individual VOCs for the tested Geneva and Weiki cultivars.

The major volatiles found in the VOCs profile of Geneva were hexanal and 1-hexanol. However, terpinolene, ethyl butanoate, and eucalyptol were the predominant volatiles found in the VOCs profile of Weiki. Moreover, Weiki manifested an almost two-fold higher concentration of volatiles than Geneva. Similarly, a one and a half-fold higher number of VOCs was identified in the untreated Weiki than in the Geneva. Terpenoids were found to be the dominant VOC group in both of the tested cultivars.

The HHP treatment led to a substantial decrease in the contribution of terpenoids, but also increased the contribution of alcohols and aldehydes to the overall VOCs profile of the processed Geneva and Weiki fruit. As shown, all the analyzed Geneva and Weiki samples showed a reduction in the sum of the major individual volatiles, regardless of the parameters of the HHP treatment. However, Geneva and Weiki exposed to 450 MPa/5 min treatment manifested the smallest loss in the sum of the individual VOCs. Moreover, the treatment of Geneva at 450 MPa/5 min significantly increased the hexanal (r = 0.782) and linalool (r = 0.806) content. On the other hand, sample exposure to 450 MPa/15 min promoted the formation of methyl butanoate, ethyl hexanoate and *cis*-geraniol, simultaneously increasing the concentration of benzaldehyde (r = 0.886). Geneva treated at 550 MPa for 5 and 15 min favored 1-hexanol (r = 0.796) and *cis*-geraniol (r = 0.582) concentrations, respectively.

For the case of Weiki, exposure to 450 MPa/5 min favored the formation of trans-2-heptenal (r = 0.999), 1-hexanol, and linalool (r = 0.970). The concentrations of (-)-terpinen-4-ol and *cis*-geraniol are also found to be highly correlated with the test treatment parameters (r = 0.848 and r = 0.694, respectively). The other important volatiles (ethyl butanoate, hexanal, 1-octen-3-ol) for the aroma of *A. arguta* were also found to be highly concentrated in the HHP-treated (450 MPa/5 or 15 min) Weiki samples. However, the concentrations of benzaldehyde (r = 0.625) and 1-hexanol (r = 0.416) were correlated only with the processing of Weiki at 650 MPa for 5 and 15 min, respectively. However, the tested processing parameters led to a distinct decrease (more than 75%, on average) in the sum of the VOCs.

Taking into consideration the obtained results, we can conclude that the processing of Geneva at 450 MPa for 5 or 15 min and Weiki at 450 MPa for 5 min is required to obtain an attractive kiwiberry-based product in terms of the composition of the volatiles, since the formation of new volatiles and an increase in the concentration of pre-existing individual major volatiles of *A. arguta*, observed regarding the tested pressurization conditions, are clearly responsible for a green, fresh, fruity, and floral aroma—as required for consumer acceptance.

## Figures and Tables

**Figure 1 molecules-27-05914-f001:**
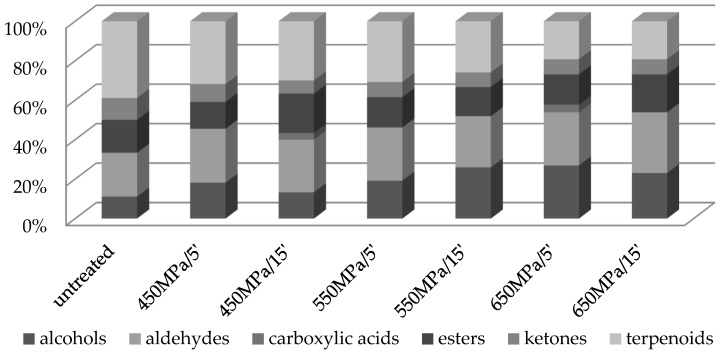
The chemical groups of VOCs identified in the untreated and the HHP-treated (450–550–650 MPa/5 and 15 min) cv Geneva.

**Figure 2 molecules-27-05914-f002:**
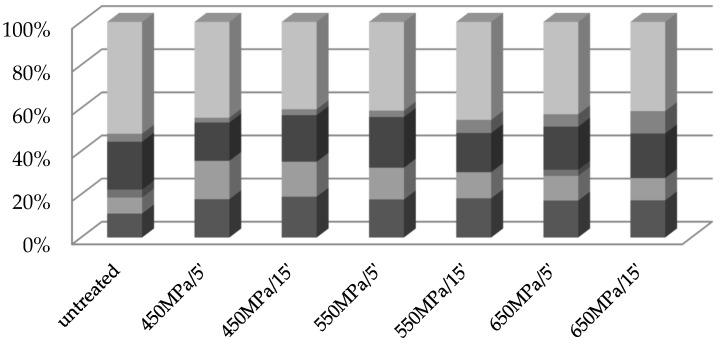
The chemical groups of VOCs identified in the untreated and the HHP-treated (450–550–650 MPa/5 and 15 min) cv Weiki.

**Figure 3 molecules-27-05914-f003:**
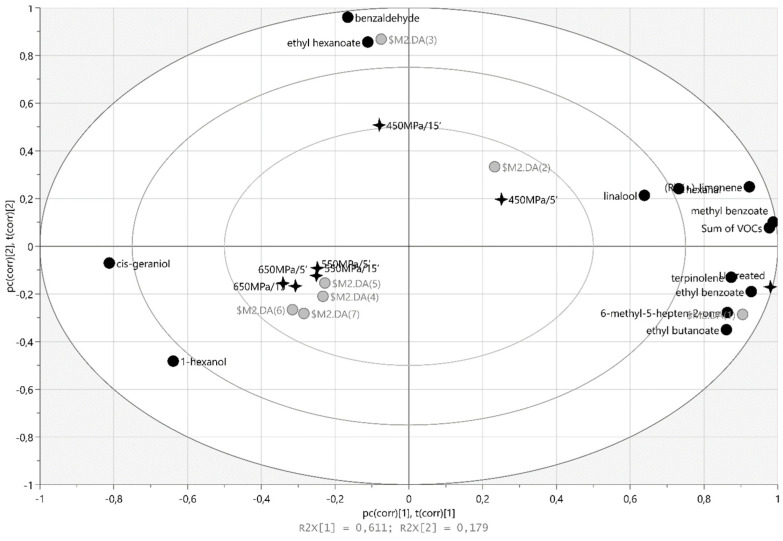
The bi-plot of PLS-DA for VOCs identified in the untreated and HHP-treated (450–550–650 MPa/5 and 15 min) cv Geneva.

**Figure 4 molecules-27-05914-f004:**
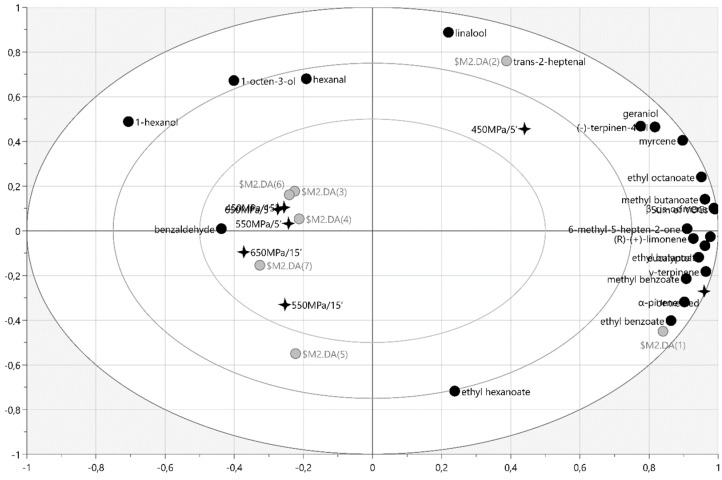
The bi-plot of PLS-DA for VOCs identified in the untreated and HHP-treated (450–550–650 MPa/5 and 15 min) cv Weiki.

**Table 1 molecules-27-05914-t001:** Volatile organic compounds (VOCs) identified in the untreated and the HHP-treated (450–550–650 MPa/5 and 15 min) *cv* Geneva. VOCs are listed according to the experimental linear retention indices (LRI _exp._).

Compounds	Aroma Description ^1^	LRI _exp._	LRI _lit._ ^2^	Untreated	450MPa/5′	450MPa/15′	550MPa/5′	550MPa/15′	650MPa/5′	650MPa/15′
methyl butanoate	fruity, apple	990	993			+				
(*Z*)-2-decanal	-	988	984			+				
ethyl butanoate	fruity, pineapple	1030	1026	+	+	+	+	+	+	+
pentanal	fermented, fruity	1088	1097							+
*α*-myrcene	sweet, fruity	1170	1176	+						
*α*-terpinene	fruity, lemon	1190	1188	+	+	+	+	+	+	+
*α*-limonene	citrus, fruity	1210	1212	+	+	+	+	+		
isopentyl alcohol	fermented	1220	1221					+	+	+
hexanal	grassy	1226	1228	+	+	+	+	+	+	+
ethyl hexanoate	fruity, strawberry	1240	1240			+	+	+	+	+
*o*-cymene	citrus	1245	1248			+				
hexyl acetate	fruity	1285	1280							+
*α*-terpinolene	sweet, fruity	1288	1297	+	+	+	+	+	+	+
2-hexenal	fruity, green	1290	1300	+	+	+	+	+	+	+
1-octen-3-one	earthy, mushroom	1315	1317	+	+	+	+	+	+	+
octanal	lemon, citrus	1320	1319		+	+	+	+	+	+
6-methyl-5-hepten-2-one	green, citrus	1330	1330	+	+	+	+	+	+	+
1-hexanol	fatty, fruity	1359	1360	+	+	+	+	+	+	+
3-hexen-1-ol	-	1377	1384							+
3-octanol	citrus, nutty	1415	1417		+	+	+	+	+	+
(*E*)-2-hexen-1-ol	vegetable	1420	1420	+	+	+	+	+	+	+
ethyl caprylate	fruity, sweet	1424	1421			+				
(*Z*)-2-heptenal	-	1435	1437	+	+	+	+	+	+	+
3,5,5-trimethyl-2-hexene	-	1452	MS ^3^		+	+	+			
*p*-*α*-dimethyl-styrene	phenolic, spicy	1439	1440		+	+	+	+	+	+
(*E*)-2-octenal	fatty, fruity	1459	1461			+	+	+	+	+
2-ethyl-1-hexanol	citrus	1480	1484					+	+	+
benzaldehyde	burnt sugar	1522	1525	+	+	+	+	+	+	+
*α*-linalool	sweet, lemon	1560	1562	+	+	+	+	+	+	+
terpinen-4-ol	sweet, fruity	1590	1593	+	+	+	+	+		
methyl benzoate	flowery, honey	1614	1615	+	+	+	+	+	+	+
(*E*,*E*)-2,4-heptadienal	-	1623	1627		+	+	+	+	+	+
butanoic acid	oily	1624	1628						+	
ethyl benzoate	*Chamomile* flower, celery, fruity	1640	1644	+	+	+	+	+	+	+
1-nonanol	floral	1662	1668		+	+	+	+	+	+
carvone	basil, bitter	1743	1749	+						
*cis*-geraniol	floral, geranium	1838	1840			+	+	+	+	+
*p*-mentha-1,8-dien-9-ol	-	1955	1962				+	+	+	+
octanoic acid	fruity, acidic	2053	2046			+				

^1^ Aroma descriptors collected from pherobase.com and pubchem.com. ^2^ Linear retention indices according to data in the literature. ^3^ MS—compounds identified according to their mass spectrum.

**Table 2 molecules-27-05914-t002:** Volatile organic compounds (VOCs) identified in the untreated and HHP-treated (450–550–650 MPa/5 and 15 min) cv Weiki. VOCs are listed according to the experimental linear retention indices (LRI _exp._).

Compounds	Aroma Description ^1^	LRI _exp._	LRI _lit._ ^2^	Untreated	450MPa/5′	450MPa/15′	550MPa/5′	550MPa/15′	650MPa/5′	650MPa/15′
pentanal	fermented, fruity	988	984		+	+	+	+		
methyl butanoate	fruity, apple	990	993	+	+	+	+			
*α*-pinene	pine	1005	1008	+	+	+	+	+	+	+
ethyl butanoate	fruity, pineapple	1030	1026	+	+	+	+	+	+	+
propionoic acid propyl ester	apple, pine	1045	1059				+			
hexanal	grassy	1088	1097	+	+	+	+	+	+	+
(Z)-2-undecane	fruity	1163	1173					+	+	
*α*-myrcene	sweet, fruity	1170	1176	+	+	+	+	+	+	+
*α*-terpinene	fruity, lemon	1190	1188		+					
2-heptanone	banana-like	1186	1185							+
*α*-limonene	citrus, fruity	1210	1212	+	+	+	+	+	+	+
(2*Z*)-3-pentyl-2,4-pentadien-1-ol	-	1218	MS ^3^		+					
isopentyl alcohol	fermented	1220	1221							+
eucalyptol	minty, sweet	1224	1224	+	+	+	+	+	+	+
2-hexenal	fruity, green	1226	1228	+	+	+	+	+	+	+
ethyl hexanoate	fruity, strawberry	1240	1240	+	+	+	+	+	+	+
*γ*-terpinene	bitter, citrus	1241	1243	+	+	+	+	+	+	
*β*-*cis*-ocimene	citrus	1243	1245	+	+					
*o*-cymene	citrus	1245	1248	+	+		+	+	+	+
*p*-cymene	lemon, fruity	1275	1277			+				
hexyl acetate	fruity	1285	1280		+	+	+	+	+	+
*α*-terpinolene	sweet, fruity	1288	1297	+	+	+	+	+	+	+
octanal	lemon, citrus	1290	1300			+	+	+	+	
3-hydroxy-2-butanone	-	1299	MS							+
2,3-dimethyl-3-undecanol	-	1305	MS			+				
(*Z*)-3-hexen-1-ol, acetate	banana, floral	1312	1313			+	+	+	+	+
1-octen-3-one	earthy, mushroom	1315	1317					+	+	
(*Z*)-2-heptenal	-	1320	1319		+					
(*E*)-2-hexen-1-ol, acetate	green	1324	1325		+	+	+	+	+	
6-methyl-5-hepten-2-one	green, citrus	1330	1330	+	+	+	+	+	+	+
*E*,*E*-2,6-dimethyl-1,3,5,7-octatetraene	-	1345	MS	+						
1-hexanol	fatty, fruity	1359	1360		+	+	+	+	+	+
2,6-dimethyl-2,6-octadiene	-	1364	MS		+	+				
3-hexen-1-ol	-	1377	1384				+	+	+	
nonanal	fruity, citrus	1412	1415			+				
3-octanol	citrus, nutty	1415	1417		+	+	+	+		
6-methyl-3-heptanol	-	1418	MS						+	
(E)-2-hexen-1-ol	vegetable	1420	1420		+	+	+	+	+	+
ethyl caprylate	fruity, sweet	1424	1421						+	+
(E)-2-octenal	fatty, fruity	1435	1437		+	+				
*p*-*α*-dimethyl-styrene	phenolic, spicy	1439	1440	+	+	+	+	+	+	+
ethyl octanoate	fruity, sweet	1440	1441	+	+					
1-octen-3-ol	mushroom-like	1445	1451	+	+	+	+	+	+	+
3,5,5-trimethyl-2-hexene	-	1452	MS		+	+	+	+	+	+
(*E*,*E*)-2,4-heptadienal	-	1459	1461		+					
2-ethyl-1-hexanol	citrus	1480	1484						+	+
benzaldehyde	burnt sugar	1522	1525						+	+
(*Z*)-2-nonenal	fatty, cucumber	1530	1528				+			
*α*-linalool	sweet, lemon	1560	1562		+	+	+	+	+	+
terpinen-4-ol	sweet, fruity	1590	1593	+	+	+	+	+	+	+
*p*-menth-1-en-4-ol	-	1608	MS			+				
methyl benzoate	flowery, honey	1614	1615	+	+	+				
(*Z*)-2-decanal	-	1623	1627		+					
butanoic acid	oily	1624	1628	+						
verbenol	fresh, pine	1625	1629		+					
ethyl benzoate	*Chamomile* flower, celery, fruity	1640	1644	+	+	+	+	+	+	+
1-nonanol	floral	1662	1668		+					
*cis*-citral	lemon-like	1674	1678		+					
*α*-terpineol	peach, fruity	1730	1720	+	+	+	+	+	+	+
verbenone	minty, spicy	1733	1735	+	+					
carvone	basil, bitter	1743	1749		+					
*cis*-geraniol	floral, geranium	1838	1840	+	+	+	+	+	+	
*cis*-carveol	citrus, fruity	1841	1846	+						
*trans*-geraniol	floral	1860	1865		+	+	+	+	+	+
*p*-cymen-8-ol	floral, sweet, citrus	1882	1887	+	+					
2-phenylethanol	rose-like, fruity	1928	1932			+	+	+		
*p*-mentha-1,8-dien-9-ol	-	1955	1962	+	+					
octanoic acid	fruity, acidic	2053	2046						+	

^1^ Aroma descriptors collected from pherobase.com and pubchem.com. ^2^ Linear retention indices according to data in the literature. ^3^ MS—compounds identified according to their mass spectrum.

**Table 3 molecules-27-05914-t003:** The content of major VOCs estimated in the untreated and HHP-treated cv Geneva. Results are presented as µg per kg of fruit.

No.	Compounds	Untreated	450MPa/5′	450MPa/15′	550MPa/5′	550MPa/15′	650MPa/5′	650MPa/15′
1	methyl butanoate	nd	nd	242.89 ± 0.89	nd	nd	nd	nd
2	ethyl butanoate	818.71 ± 5.19 ^a^	428.33 ± 25.00 ^b^	179.98 ± 44.62 ^d^	196.24 ± 3.05 ^d^	466.72 ± 20.21 ^b^	274.81 ± 11.95 ^c^	291.50 ± 16.87 ^c^
3	myrcene	122.42 ± 10.93	nd	nd	nd	nd	nd	nd
4	(R)-(+)-limonene	266.80 ± 29.15 ^a^	157.48 ± 3.22 ^bc^	174.05 ± 16.66 ^b^	128.03 ± 8.43 ^bcd^	104.32 ± 56.45 ^cd^	67.05 ± 1.48 ^d^	71.71 ± 0.91 ^d^
5	hexanal	1183.20 ± 30.68 ^b^	1645.28 ± 168.09 ^a^	521.37 ± 61.17 ^c^	658.67 ± 50.25 ^c^	211.62 ± 13.92 ^d^	271.54 ± 12.19 ^d^	230.69 ± 17.91 ^d^
6	ethyl hexanoate	nd	nd	311.98 ± 29.16 ^a^	nd	45.81 ± 1.83 ^b^	nd	nd
7	terpinolene	819.93 ± 36.50 ^a^	93.83 ± 5.50 ^d^	277.44 ± 21.60 ^b^	146.14 ± 4.81 ^c^	138.62 ± 17.08 ^c^	152.54 ± 4.55 ^c^	113.96 ± 11.66 ^cd^
8	6-methyl-5-hepten-2-one	614.33 ± 59.1 ^a^	123.84 ± 12.32 ^c^	164.64 ± 20.35 ^bc^	114.13 ± 17.49 ^c^	222.65 ± 9.71 ^b^	114.17 ± 2.33 ^c^	177.50 ± 25.64 ^bc^
9	1-hexanol	831.93 ± 19.15 ^f^	803.75 ± 10.89 ^f^	886.68 ± 5.19 ^e^	1515.37 ± 14.20 ^a^	1132.90 ± 24.08 ^c^	1179.57 ± 5.91 ^b^	1094.46 ± 7.58 ^d^
10	benzaldehyde	89.94 ± 12.19 ^f^	284.82 ± 17.81 ^b^	527.87 ± 17.88 ^a^	191.69 ± 4.96 ^d^	240.14 ± 2.75 ^c^	116.89 ± 10.44 ^ef^	129.24 ± 4.42 ^e^
11	linalool	212.38 ± 20.92 ^b^	264.00 ± 9.57 ^a^	159.65 ± 11.84 ^de^	157.63 ± 9.67 ^de^	127.41 ± 14.72 ^c^	179.26 ± 5.26 ^e^	134.22 ± 1.35 ^cd^
12	(-)-terpinen-4-ol	518.74 ± 34.05	nd	nd	nd	nd	nd	nd
13	methyl benzoate	406.24 ± 39.40 ^a^	224.82 ± 14.86 ^b^	178.53 ± 13.49 ^c^	100.27 ± 1.37 ^de^	129.06 ± 4.32 ^d^	68.12 ± 8.65 ^e^	75.74 ± 5.56 ^e^
14	ethyl benzoate	608.44 ± 46.35 ^a^	138.35 ± 5.47 ^b^	155.66 ± 15.77 ^b^	134.91 ± 6.47 ^b^	132.21 ± 3.96 ^b^	61.11 ± 5.36 ^c^	123.92 ± 5.63 ^b^
15	*cis*-geraniol	nd	nd	44.31 ± 4.15 ^bc^	36.14 ± 1.70 ^c^	68.18 ± 13.15 ^a^	51.62 ± 1.98 ^b^	40.24 ± 3.07 ^bc^
	*Sum of VOCs*	6493.06 ± 343.69 ^a^	4164.5 ± 272.73 ^b^	3825.05 ± 262.77 ^bc^	3379.22 ± 122.4 ^cd^	3019.64 ± 182.18 ^de^	2536.68 ± 70.1 ^ef^	2483.18 ± 100.6 ^f^

Data expressed as mean ± standard deviation (*n* = 3); ^a–f^—different letters within the same row indicate statistically significant differences at *p* < 0.05 using one-way ANOVA, followed by Tukey’s test.

**Table 4 molecules-27-05914-t004:** The content of major VOCs estimated in the untreated and HHP-treated cv Weiki. Results are presented as µg per kg of fruit.

No.	Compounds	Untreated	450MPa/5′	450MPa/15′	550MPa/5′	550MPa/15′	650MPa/5′	650MPa/15′
1	methyl butanoate	127.11 ± 13.32 ^a^	89.53 ± 8.30 ^b^	31.65 ± 0.42 ^c^	34.53 ± 0.55 ^c^	nd	nd	nd
2	*α*-pinene	146.75 ± 12.27 ^a^	58.68 ± 7.55 ^b^	42.25 ± 1.99 ^bc^	51.48 ± 0.21 ^b^	46.32 ± 3.85 ^bc^	53.22 ± 6.51 ^b^	30.04 ± 3.75 ^c^
3	ethyl butanoate	2324.55 ± 29.85 ^a^	1275.02 ± 37.12 ^b^	182.96 ± 12.37 ^e^	269.39 ± 35.13 ^d^	99.58 ± 10.02 ^f^	119.80 ± 4.77 ^ef^	558.99 ± 43.29 ^c^
4	hexanal	340.77 ± 4.67 ^d^	447.49 ± 9.16 ^bc^	588.37 ± 26.53 ^a^	486.03 ± 66.48 ^b^	188.87 ± 6.05 ^e^	475.37 ± 3.08 ^bc^	402.83 ± 13.91 ^cd^
5	myrcene	1004.91 ± 20.34 ^a^	988.95 ± 109.63 ^a^	360.49 ± 26.38 ^b^	422.48 ± 15.60 ^b^	24.00 ± 3.33 ^d^	379.08 ± 25.65 ^b^	229.97 ± 23.68 ^c^
6	(*R*)-(+)-limonene	713.78 ± 5.32 ^a^	590.85 ± 80.57 ^b^	114.24 ± 11.83 ^e^	131.09 ± 4.47 ^e^	364.47 ± 6.28 ^c^	243.97 ± 19.03 ^d^	122.71 ± 16.77 ^e^
7	eucalyptol	2009.82 ± 91.88 ^a^	777.48 ± 58.24 ^b^	444.37 ± 40.60 ^c^	399.41 ± 29.66 ^c^	111.80 ± 1.83 ^d^	449.25 ± 10.87 ^c^	49.79 ± 2.53 ^e^
8	ethyl hexanoate	368.49 ± 37.52 ^b^	88.00 ± 6.86 ^c^	20.97 ± 1.24 ^d^	19.03 ± 1.89 ^d^	661.31 ± 11.37 ^a^	23.13 ± 3.42 ^d^	84.97 ± 1.43 ^c^
9	*γ*-terpinene	246.41 ± 36.62 ^a^	87.10 ± 2.58 ^b^	24.40 ± 2.34 ^c^	31.51 ± 1.59 ^c^	14.26 ± 0.72 ^c^	26.44 ± 1.51 ^c^	nd
10	*β*-*cis*-ocimene	72.05 ± 2.07 ^a^	53.21 ± 2.25 ^b^	nd	nd	nd	nd	nd
11	terpinolene	2908.45 ± 25.27 ^a^	1607.05 ± 28.69 ^b^	349.31 ± 37.71 ^e^	487.07 ± 18.14 ^cd^	110.51 ± 6.46 ^f^	436.15 ± 10.01 ^d^	543.18 ± 41.67 ^c^
12	trans-2-heptenal	nd	72.99 ± 2.42	nd	nd	nd	nd	nd
13	6-methyl-5-hepten-2-one	185.79 ± 4.95 ^a^	96.56 ± 2.42 ^b^	66.88 ± 0.96 ^d^	44.61 ± 0.81 ^e^	19.28 ± 1.19 ^f^	81.82 ± 5.00 ^c^	36.92 ± 0.69 ^e^
14	1-hexanol	nd	161.23 ± 23.24 ^c^	295.00 ± 16.40 ^a^	223.80 ± 5.32 ^b^	48.94 ± 1.54 ^d^	277.19 ± 6.25 ^a^	301.71 ± 18.66 ^a^
15	ethyl octanoate	260.52 ± 18.78 ^a^	259.48 ± 5.33 ^a^	nd	nd	nd	nd	nd
16	1-octen-3-ol	40.40 ± 0.86 ^d^	86.15 ± 11.26 ^b^	100.12 ± 5.47 ^ab^	85.95 ± 3.63 ^b^	60.12 ± 1.69 ^c^	107.53 ± 4.86 ^a^	46.46 ± 3.93 ^cd^
17	benzaldehyde	nd	nd	nd	nd	nd	12.05 ± 1.22 ^a^	11.53 ± 0.25 ^a^
18	linalool	nd	595.66 ± 54.21 ^a^	113.54 ± 13.02 ^b^	98.65 ± 6.71 ^b^	19.53 ± 2.24 ^c^	133.66 ± 2.78 ^b^	89.96 ± 7.83 ^b^
19	(-)-terpinen-4-ol	784.89 ± 78.01 ^b^	1328.86 ± 212.42 ^a^	68.47 ± 6.81 ^c^	49.57 ± 3.18 ^c^	101.47 ± 13.17 ^c^	68.00 ± 4.34 ^c^	12.25 ± 1.24 ^c^
20	methyl benzoate	90.04 ± 5.71 ^a^	22.71 ± 3.38 ^b^	24.05 ± 2.68 ^b^	nd	nd	nd	nd
21	ethyl benzoate	280.35 ± 6.15 ^a^	38.36 ± 0.13 ^b^	28.42 ± 3.78 ^c^	28.95 ± 0.75 ^c^	21.66 ± 3.23 ^c^	29.90 ± 2.24 ^bc^	26.63 ± 2.95 ^c^
22	*cis*-geraniol	33.64 ± 2.19 ^b^	43.36 ± 0.45 ^a^	16.55 ± 1.59 ^d^	16.05 ± 0.89 ^d^	14.56 ± 0.31 ^d^	23.80 ± 2.04 ^c^	nd
	*Sum of VOCs*	11938.72 ± 395.78 ^a^	8768.72 ± 666.21 ^b^	2872.04 ± 212.12 ^c^	2879.60 ± 195.01 ^c^	1906.68 ± 73.28 ^d^	2940.36 ± 113.58 ^c^	2547.94 ± 182.58 ^cd^

Data expressed as mean ± standard deviation (n = 3); ^a–f^—different letters within the same row indicate statistically significant differences at *p* < 0.05 using one-way ANOVA, followed by Tukey’s test.

## Data Availability

The data are available upon request.

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
