# Peer review of "Characterization of Volatile Organic Compounds in Kiwiberries (Actinidia arguta) Exposed to High Hydrostatic Pressure Processing by HS-SPME/GC-MS"

_molecules, 2022, doi:10.3390/molecules27185914_

Round 1

Reviewer 1 Report

The authors Characterized the profile of volatile organic compounds in kiwiberry cultivars exposed to HHP. Although the findings are interesting, I have some questions as follows.

 1. Line 14-17 in the abstract (“The sum of individual VOCs……”):  The data showed at the end of table 3 and 4. The authors describe in the abstract but didn’t discussion in the results and discussion part.  The authors should provide more information or discussion about the reduction of the sum of VOCs after treated with HHP. Is the reduction of the sum of VOCs affects to the quality of kiwiberry products?

2. “ A. arguta” at line 124 and 133 should be italic

3. Table 3 and 4 : The authors need to describe statistic in the footnote.

Author Response

The answers to the Reviewer's comments are attached in the file.

Reviewer 2 Report

The authors succeeded to profile volatile organic compounds (VOCs) in kiwiberis as a results of high hydrostatic pressure (HHP) processing. The classification of VOCs based on chemical groups is very informative and convenient for the readers to comprehend the study with big data. I have a few suggestions to improve the manuscipt for publications as per followings

1) The study could have been improved if the effect of HHP on major enzyme inactivation such as polyphenol oxidase (PPO) was highlighted. 

2) To indicate clearly the form of kiwiberries after HHP. Is it still in fruit form or paste or concentrate. we are well aware that the HHP conditions are extremely severe that it could change the form of the fruits.

3) To make it clear why such condition of HHP is used. Is it for preservation purposes?

4) The reporting should be in passive form and past tense.

5) The and LRI multivariate PLS-DA could be explained more clearly in the methodology as how it was determined and analysed.

Author Response

(The authors gave the same response as above.)

Reviewer 3 Report

The present work deals with the characterization of volatile organic compounds (VOCs) in kiwiberries

(Actinidia arguta) exposed to high-hydrostatic pressure processing by HS-SPME/GC-MS. The major volatiles found in the VOC profile of ‘Geneva’ are hexanal and 1-hexanol, whereas terpinolene, ethyl butanoate and eucalyptol were the predominant volatiles found in the VOC profile of ‘Weiki’. Terpenoids were found to be the dominant VOC group in both of the tested cultivars. From the results achieved it was demonstrated that the processing of ‘Geneva’ at 450 MPa for 5 or 15 min and ‘Weiki’ at 450 MPa for 5 min is required to obtain an attractive kiwiberry-based product in terms of the composition of the volatiles. 

The work is interesting and the novelty of the work is clearly stated. In fact, till date, there are only scarce data available regarding the effect of pressurization on the profile of volatile compounds in kiwiberries. 

  • I invite the authors to revise English. Past tense form must be used throughout the whole manuscript. 
  • Superimposed GC chromatograms related to the VOCs identified in the untreated and the HHP-treated 'Geneva' and ‘Welki’ cultivars with proper peak identification must be provided. 
  • Section 2.2. Quantitative analysis of VOCs in ‘Geneva’ and ‘Weiki’ cultivars. The authors must discuss how quantitative results were achieved. It is understood that GC-MS equipment was employed. In section 3.5: Gas chromatography-mass spectrometry (GC-MS) analysis, they refer to the use of calibration curves for the external standards listed in section 3.1 despite no other information was provided. 

Author Response

The answers to the Reviewer's comments please find in the attached file.

Round 2

Reviewer 3 Report

The authors have adeqautely addressed all remarks. The improved manuscript can be now accepted in the present form.